# Environmentally Safe Biosynthesis of Gold Nanoparticles Using Plant Water Extracts

**DOI:** 10.3390/nano11082033

**Published:** 2021-08-10

**Authors:** Mohadeseh Hassanisaadi, Gholam Hosein Shahidi Bonjar, Abbas Rahdar, Sadanand Pandey, Akbar Hosseinipour, Roohollah Abdolshahi

**Affiliations:** 1Department of Plant Protection, Shahid Bahonar University of Kerman, Kerman 7618411764, Iran; mhassani@agr.uk.ac.ir (M.H.); Hosseini@uk.ac.ir (A.H.); 2Medical Mycology and Bacteriology Research Center, Kerman University of Medical Sciences, Kerman 7616913555, Iran; 3Department of Physics, Faculty of Science, University of Zabol, Zabol 98615-538, Iran; a.rahdar@uoz.ac.ir; 4Department of Chemistry, College of Natural Science, Yeungnam University, 280 Daehak-ro, Gyeongsan 38541, Korea; 5Department of Agronomy and Plant Breeding, Shahid Bahonar University of Kerman, Kerman 7618411764, Iran; abdoshahi@uk.ac.ir

**Keywords:** gold nanoparticle, green synthesis, medicinal plant, folkloric medicine, water extract, Middle East

## Abstract

Due to their simplicity of synthesis, stability, and functionalization, low toxicity, and ease of detection, gold nanoparticles (AuNPs) are a natural choice for biomedical applications. AuNPs’ unique optoelectronic features have subsequently been investigated and used in high-tech applications such as organic photovoltaics, sensory probes, therapeutic agents, the administration of drugs in biological and medical applications, electronic devices, catalysis, etc. Researchers have demonstrated the biosynthesis of AuNPs using plants. The present study evaluates 109 plant species used in the traditional medicine of Middle East countries as new sources of AuNPs in a wide variety of laboratory environments. In this study, dried samples of bark, bulb, flower, fruit, gum, leaf, petiole, rhizome, root, seed, stamen, and above-ground parts were evaluated in water extracts. About 117 plant parts were screened from 109 species in 54 plant families, with 102 extracts demonstrating a bioreduction of Au^3+^ to Au^0^, revealing 37 new plant species in this regard. The color change of biosynthesized AuNPs to gray, violet, or red was confirmed by UV-Visible spectroscopy, TEM, FSEM, DLS, and EDAX of six plants. In this study, AuNPs of various sizes were measured from 27 to 107 nm. This study also includes an evaluation of the potency of traditional East Asian medicinal plants used in this biosynthesis of AuNPs. An environmentally safe procedure such as this could act as a foundation for cosmetic industries whose quality assessment systems give a high priority to non-chemically synthesized products. It is crucial that future optimizations are adequately documented to scale up the described process.

## 1. Introduction

As defined by the European Commission (EC), nanomaterials are particles that have at least 50% of their number, which is measured in one or more external dimensions, with a size distribution of less than 100 nm. In other words, nanomaterials are particles of one or more dimensions of 1–100 nm [1]. Green nano-biotechnology aims to construct nanoparticles (NPs) in environmentally safe methods. This approach has attracted the interest in researchers in many related fields such as medicine, biology, and microbiology [2,3,4]. Among NPs, metallic NPs (MNPs) such as Au, Ag, Zn, Cu, etc. have been widely studied by many researchers for their phenomenal properties [5,6,7,8]. Gold nanoparticles (AuNPs), among these, are an exciting topic to research because of their stability and biocompatibility, oxidation resistance [9], Surface Plasmon Resonance (SPR) [10,11], low toxic activities [12], and application in medicine and agriculture in drug and agrochemical delivery systems [13,14], sensory probes [15], and diagnostic tools [16,17]. The SPR is a unique behavior in AuNPs that has made them valuable in the medical and biological sciences such as in cancer nanotherapy [18], X-ray Computed Tomography (CT), and magnetic resonance imaging (MRI) [19]. In addition, the electrokinetic behavior of the AuNPs has caused it to be a valuable formulation for cosmetic purposes [20]. Numerous techniques, including chemical, physical, and biological, are used to produce NPs [21]. The chemical and physical methods have been reported to have adverse environmental impacts and harmful effects on human health [22]. As a result of the non-safe and detrimental consequences of the methods mentioned above, the biosynthesis of NPs by many biological agents including plant (phytonanotechnology) [23,24], bacteria [25], fungi [26], and algae [27] as a nontoxic, simple, and cost-effective method is taken into consideration as greener syntheses [28,29]. By reducing biomolecules found in these organisms, they convert metal ions into MNPs [30]. The easy availability, cultivability, and possibility of the large-scale production potential of plants makes them the preferred candidates in nano field studies [31,32]. Various bioactive molecules found in plants, including proteins, carbohydrates, organic acids, vitamins, alkaloids, and secondary metabolites have been reported to act as bioreductive, capping, and stabilizing agents for chloride and nitrate precursors within the biosynthesis of specific NPs [33,34]. For example, several alkaloids in *Areca catechu* such as arecoline, arecaidine, arecolidine, guvacine, guvacoline, isoguvacine, norarecaidine, and norarecoline belong to the pyridine group and play a prominent role in this reduction process [35]. All of these bioactive molecules can cause a reduction of Au^3+^ in the process of biosynthesis of nanoparticles. The extraction of most plant parts, including leaves, flowers, undergrounds (root), and seeds, can act as regenerative agents [36]. Since ancient times, medicinal plants in the Middle East countries have been widely used [37]. These plants are crucial resources in folkloric medicine and provide valuable capital for modern medical science to develop natural drugs; however, investigating medicinal plants would help discover their hidden and phenomenal potentials [38,39]. Their use in ancient remedies of diseases [40] and their application to biosynthesize NPs would expand their bioactivity beyond known boundaries. In such a school of thought, green nanomaterials may manifest as new natural bioresources to be the platforms in many scopes of nanotechnology. In recent years, there has been a rising awareness of the discovery of the potential of plants as major bioresources for NPs biosynthesis and their application in phytonanotechnology as AuNPs [41,42]. The biosynthesis of AuNPs via phytonanotechnology approaches is supported as an uncomplicated, rapid, and affordable method and is considered a biocompatible, nontoxic method with a large-scale application [24].

Historically, for the first time, Gardea-Torresdey et al. [43] reported the capability of plants to produce AuNPs. They used live alfalfa (*Medicago sativa*) as the reducing agent for AuNPs biosynthesis. Subsequently, many scientists investigated the ability of plant extracts for bioreduction AuNPs. These assessments are documented by several reviews [44,45,46,47]. In the most recent reports, several investigators showed the biosynthesis of AuNPs using *Limnophilarugosa* [48], *Linumusitatissimum* [49], *Opuntiapycnacantha* [42], *Nigella sativa* [50], *Nothapodytesfoetida* [51], *Persicariasalicifolia* [52], *Garcinia kola* [53], Citrus aurantifulia [54], *Salvadorapersica* [55], and *Hibiscus syriacus* [56]. Karmous et al. 2020 studied the feasibility and advantages of plant-based synthesized NPs in the prevention, diagnosis, and therapy of cancer [57,58]. Interestingly, they revealed the mechanism by which their plant-based NPs interacted with constituents of cancerous cells. Authors concluded that green NPs can act as novel tools for prognostic biomarkers in diagnosis of cancer and targeted drug delivery in tumor cells. They also expressed that biosynthesized NPs either reach the damaged tumor cells or protect healthy cells via the antioxidative and antitumor agents found in plants.

In the present survey, we report 27 medicinal plant species (in 18 families) of the Middle East to be new bioresources in the biosynthesis of AuNPs. This assessment was performed using plant water extracts in a simple, green, and cost-effective procedure. Despite the progressive growth of reports on the introduction of potent plants in the biosynthesis of AuNPs, there is a wide range of unknown valuable medicinal plants that have not yet been evaluated for the biosynthesis of AuNPs. To minimize such a gap of information, we aimed to (1) Screen common medicinal plants, which can mediate the biosynthesis of AuNPs; (2) Evaluate 117 plant parts from 109 plant species of 54 families for their ability to reduce Au^3+^ to Au^0^; and (3) Perform an instrumental characterization of six biosynthesized samples of AuNPs by UV-Visible spectroscopy, TEM, FSEM, DLS, and EDAX.

## 2. Materials and Methods

### 2.1. Collection of Plant Materials and Preparation of Water Extracts

The 117 healthy and dried plant parts of 109 species were obtained from the Laboratory of Plant Systematics, College of Agriculture, Shahid Bahonar University of Kerman, Iran to prepare herbal water extracts. The plant samples were pulverized using a mortar and pestle. Each plant powder was soaked in distilled water at a 1:100 ratio (*w*/*v*), shaken continuously for 5 min, and kept at ambient temperature. After 12 h, plant water extracts were filtered through Whatman filter paper No. 1. For more clarity, filtrates were centrifuged using a low-speed bench centrifuge at 5000 rpm for 20 min. The supernatant samples were collected and refrigerated before use.

### 2.2. Biosynthesis of AuNPs

The biosynthesis of AuNPs was performed according to Guo et al. [59]. Tetrachloroauric acid (HAuCl4) was purchased from Sigma-Aldrich (Saint Louis, Missouri, USA). A 0.1 M solution of AuNPs was prepared in deionized water (DW). For the reduction of Au^3+^ ions to Au^0^, the water extracts were used as regenerative agents. The purified plant extracts were mixed with 2 mL of 0.1% HAuCl4 solution to produce Au^3+^ ions concentrations of 0.05 M, and the solution was incubated overnight at ambient temperature. A continuous shaking of the tubes was performed during this time in order to speed up biosynthesis and prevent aggregation. A control treatment consisted of mixing 2 mL of each water plant extract with 2 mL of deionized water and treating as described above. A biosynthesis of AuNPs was examined 20–24 h after the mixtures were formulated.

### 2.3. Instrumentation Analyses of AuNPs

#### 2.3.1. Visual Color Grading

The first indicator for the biosynthesis of NPs is the color change of the mixture solutions [60]. In the process of biosynthesis of AuNPs, the regeneration of the Au^3+^ to Au^0^ causes a color change from yellow or colorless to gray, violet, or red. The color changes of reaction mixtures occur following the collective oscillations of electrons on the surface of AuNPs [61]. In order to rank each sample according to the degree of color change, colors were compared to a control and rated from 0 to 4, which indicates no, slight, moderate, intense, and very drastic color changes, respectively.

#### 2.3.2. Selected Samples for Instrumental Analysis

Six plant samples were selected for further laboratory testing from vast samples that indicated color changes. The selected criteria included red to violet color intensity, better clarity and least turbidity, minimum sedimentation, and low aggregation. The selected plants include *Rosa damascena* (Rosaceae), *Juglans regia* (Juglandaceae), *Urticadioica* (Urticaceae), *Areca catechu* (Arecaceae), *Caccinia macranthera* (Boraginaceae), and *Anethum graveolens* (Apiaceae).

#### 2.3.3. UV-Visible Spectroscopy

The UV-visible spectroscopic analysis can verify the bioreduction of Au^3+^ to Au^0^ accurately [23]. An AuNP’s absorbance spectrum ranges from 350 to 600 nm, which is determined by its morphological properties, distribution, and optical properties [61]. This experiment involved analyzing selected extracts at room temperature with a UV-Vis spectrophotometer (Varian Cary 50 Bio UV/Visible Spectrophotometer). A blank was prepared in all samples using water extracts of relevant plants without Au ions.

#### 2.3.4. TEM Analyses

To understand the morphology of the biosynthesized colloidal AuNPs, transmission electron microscopy (TEM) was conducted using an LEO 912 AB TEM operating at an accelerating voltage of 30 to 100 kV. Each sample was diluted with deionized water prior to testing. Then, a drop of each was applied to the carbon-coated copper grids. After 2 min, the excess of each solution sample was removed by absorbing it to the edge of a filter paper and kept for air drying. For each sample, several electron micrographs were taken.

#### 2.3.5. FSEM Analyses

Field emission scanning electron microscopy (FSEM) was adapted to facilitate the study of the topography and geometry of AuNPs. FSEM analysis is a standard method for the surface and morphological characterization of NPs [62]. Biosynthesized AuNPs were evaluated by an FSEM (FESEM TESCAN MIRA 3, Czech), and appropriate electron micrographs were prepared for all six samples. To prepare the specimens for imaging, a small quantity of each colloidal AuNPs was deposited on separate glass coverslips and allowed to dry at the ambient temperature. Then, coverslips were mounted on aluminum sputters and were examined by FSEM. Several FESEM electron micrographs were taken for each sample. Applied magnification and voltage were implanted on the corresponding electron micrographs.

#### 2.3.6. DLS Analyses

Dynamic light scattering (DLS) is a precise, standardized technique for investigating size in NPs. It provides insight into information on the means of particle sizes and particle size distribution [63]. A DLS analyzer (HORIBA SZ-100, HORIBA, Japan) was applied to measure the size of biosynthesized AuNPs for all six samples. In addition to particle size distribution, DLS provides insight into the synthesized nanoparticles’ polydispersity index value (PI). PI is a dimensionless index that exhibits the uniformity or non-uniformity of the NPs ranging from 0 to 1. Values greater than 0.7 indicate a broad particle size distribution and their high polydispersity, while values less than 0.05 indicate the high monodispersity of NPs [64,65].

#### 2.3.7. EDAX Analyses

Energy-dispersive X-ray analysis (EDAX) is a practical method for detecting the materials’ spectra. This technique is a key analysis in nano-based research to confirm the presence of the relevant element [66]. EDAX detects the elements based on characteristic emitted X-rays by the sample atoms via the incident beam electrons [67]. In this regard, EDAX elemental analysis of six biosynthesized AuNPs was performed using a field emission scanning electron microscope (FESEM TESCAN MIRA 3, Czech).

## 3. Results

### 3.1. Screened Plants

Relevant information for the evaluated medicinal plants, including their scientific, family names, voucher numbers, and parts of assayed plants are presented in Appendix A. Furthermore, the resultant colors of reaction mixtures and the color intensity of biosynthesized AuNPs of the tested plants have been documented for each specimen. Based on visual analyses, the color intensity of reaction mixtures was rated in four scores (from 0 to 4), which represent no, slight, moderate, intense, and very intense color changes, respectively.

### 3.2. Instrumentation Analyses of AuNPs

#### 3.2.1. Visual Color Grading

Visual color grading was performed by comparing biosynthesized AuNPs samples with blanks (plant extracts without Au^3+^). The sample plants that resulted in the biosynthesis of the AuNPs are presented in Appendix A. The color change due to the reduction of Au^3+^ to Au^0^ includes spectra of no color, pale-yellow, gray, violet, and red.

#### 3.2.2. UV Visible Spectroscopy

UV-Vis spectra indicated preliminary confirmation of biosynthesized AuNPs. The sharp peaks were observed at 525–545 nm. A peak of absorption was observed for the six AuNPs related to bioactive samples (Figure 1), whereas there was none for the blanks.

#### 3.2.3. TEM Analyses

TEM analysis confirmed the presence of AuNPs. As illustrated by the digital electron micrograph analysis of TEM images of six biosynthesized AuNPs via six medicinal bioactive plants, AuNPs resembled amorphous shapes (Figure 2).

#### 3.2.4. FSEM Analyses

FSEM analysis images confirmed the formation of colloidal AuNPs. The SEM electron micrograph images of six AuNPs biosynthesized by six water extracts of medicinal plants are indicated in Figure 3.

#### 3.2.5. Particle Size Distribution

Particle size distribution histograms of the AuNPs were determined from FSEM electron micrographs with the Sigma Scan Pro software (SPSS Inc., Version 4.01.003). In the present study, the average particle size range of biosynthesized AuNPs is shown in Figure 4.

#### 3.2.6. DLS Analyses

The DLS results of analyzed samples revealed that the diameters of distribution of particles sizes are in the range of 27.2 to 107.6 nm (Figure 5). However, the water layer surrounding NPs results in the detection of particles in a larger size [68,69]. Of the six samples analyzed, the smallest particle size belonged to *Juglansregia* and the largest belonged to *Urticadioica* with the average size of ≈27 and ≈107 nm, respectively. Additionally, PI values indicate acceptance dispersity (less than 0.7) for the samples [64].

#### 3.2.7. EDAX Analyses

Biosynthesized samples were confirmed to contain Au elements through EDAX analyses (Figure 6). Each sample is represented in a table on the related diagram indicating its detected elements. Note that some of the elements that occur in plant extracts are derived from other molecules.

#### 3.2.8. New Bioactive Plants

Among the 109 species tested, 27 species are described for the first time according to literature surveys, as shown in Table 1.

## 4. Discussion

Nano-based science, nanotechnology, is a rapidly growing interdisciplinary field that has attracted the attention of numerous scientific disciplines because of its widespread use in diverse areas.

The synthesis of AuNPs is achieved through chemical, physical, and biological approaches. High energy demand, high cost, the difficulty of processing, and the use of toxic reagents have led to fundamental limitations in non-biological methods [70]. In contrast, the high speed, cost-effectiveness, and environmental compatibility of biological methods have led to greater acceptance of these methods. In addition, in biological methods, the bioreagents by capping the NPs play a central role in stabilization and non-aggregation [71]. Among these, phytonanotechnology, which is based on using plants as the regenerative agents for the biosynthesis of AuNPs, is the faster, easier, and cheaper technique compared with the biosynthesis of AuNPs using other bioresources such as bacteria, fungi, and algae [72]. In this context, medicinal plants play a valuable investment role due to their bioactive compounds [73]. Recent works have documented the phytochemical importance of medicinal plants and their bioactive compounds [74,75,76,77]. The biological characteristics of medicinal plants play a pivotal role in the characterization of NPs [78,79]. The medicinal plants have bioactive molecules for use in folkloric medicine and are considered an investment for the biosynthesis of MNPs, such as AuNPs. For instance, the *Hygrophila auriculata* plant as a medicinal plant in Asia and Africa has been shown to have bioactivity in the regeneration of Au^3+^ to Au^0^ [80]. Uzma et al. [81] biosynthesized AuNPs using *Commiphora wightii* and proved their anticancer effect on breast cancer. Similarly, *Thymus vulgaris* was an efficient plant source to produce AuNPs [82]. We would like to express that special attention must be devoted to the choice of plant species for the toxic phytocompounds present in some plants, as we noticed the presence of cobalt with *Rosa*
*damascena*, iron with *Juglans regia*, and calcium with *Anethum graveolens*, *Juglans regia*, and *Caccinia macranthera* (Figure 6). However, special attention should be paid to the increase in metal or release of trace elements from metal oxide and metal biosynthesized NPs, which can result in elevated oxidative stress that is related to a higher risk of cancer onset. In other words, the selection of the plant species, organs, and extracts is very crucial, since some plants can be toxic to healthy cells. There are several reports on the beneficial action of AuNPs to treat cancers [83,84,85]. An interesting criterion of green NPs is that their surfaces selectively adsorb biomolecules while coming into contact with complex biological fluids, progressively forming a halo circumference or corona that reciprocally interacts with biological systems. These circumference layers bear additional efficacy over nude biological NPs [86]. Hence, biological NPs are more efficient due to the association of bioactive ingredients on the surface of biosynthesized NPs from the biological origins, such as micro-organisms and plants. Especially in folkloric medicinal plants, there exist vast metabolites with pharmacological activities that are believed to bind to the biosynthesized NPs, exhibiting further benefit by promoting the efficacies of the NPs [24]. At present, in spite of advancements and the curative action of AuNPs, implemented nanomaterial cannot be eliminated from patients’ blood. Shahidi Bonjar in 2013 [87] made an experimental recommendation by describing a ‘‘Nanogold detoxifying machine’’ for the filtration of idle AuNPs from the blood of treated cancer patients. The device similar to a ‘‘hemodialysis machine’’ would help to increase safety in AuNPs therapy of some cancers and prevent the accumulation of AuNPs in non-target tissues or organs after therapy. It has been reported that the main mechanism of biosynthesis of NPs by plants can be related to the reduction of ions by biomolecules such as organic acids, proteins, amino acids, vitamins, and secondary metabolites [24]. Through capping, these reducing agents enhance the colloidal stability of NPs and prevent aggregation [88].

An approach using phytonanotechnology to synthesize AuNPs led us to understand the bioactivity of many medicinal plants. This approach is fast, easy, environmentally friendly, and inexpensive. Through this biosynthetic process, we screened 117 plant parts of 109 species of medicinal plants for their capabilities in the biosynthesis of AuNPs.

It was determined that among the 117 plants used in folkloric medicine of Middle East countries, 102 plants were able to produce AuNPs. Twenty-seven plant species were reported as new bioresources in the biosynthesis of AuNPs. However, no inference can be drawn about these plants’ biological activity and curative behavior in traditional medicine. In the present research, 11 species of 12 Lamiaceae (11 out of 12), 8 Leguminosae (8 out of 9), 8 Asteraceae (8 out of 8), and 7 Apiaceae (7 out of 8) showed the most bioactivity in the biosynthesis of AuNPs.

According to our results, medicinal plants can be considered biological candidates for future applications in nanomaterials’ biosynthesis and more agricultural and medical applications. We should conduct more research to get a better understanding of nanomaterials’ environmental effects in order to accomplish this goal. Indeed, the assessment of antifungal and antibacterial activities of AuNPs against plant and human pathogens would further add to the knowledge on the efficiency of AuNPs in different sciences and the bioactivity of medicinal plants in biological processes. Evaluation of the therapeutic and diagnostic effectiveness of these biosynthesized AuNPs can help improve our knowledge of their future uses.

## 5. Conclusions

This study (a): screened 117 medicinal plant parts for the biosynthesis of AuNPs, (b) ranked plants’ performances according to their bioactivity in related families, (c) found that 102 plant parts could reduce Au^3+^ to Au^0^, (d) noticed 27 plant species are reported for the first time, and (e) revealed that the most bioactive species belong to Lamiaceae, Leguminosae, Asteraceae, and Apiaceae respectively. We believe that their mechanism for bioactivity should be thoroughly investigated. Finally, based on the spectacular achievement of nanotechnology and wide application of AuNPs in various fields, it can be concluded that the synthesis of AuNPs in line with phytonanotech goals will be a launching pad for science in the future.

## Figures and Tables

**Figure 1 nanomaterials-11-02033-f001:**
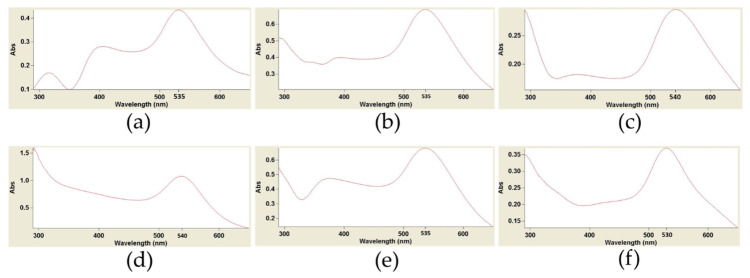
UV-Visible spectra of biosynthesized AuNPs by water extracts of six medicinal plants used in folkloric medicine of Middle East countries. Each spectrum was presented for its related sample. (**a**–**f**) indicate UV-Visible spectra of *Rosa damascena*, *Juglans regia*, *Urtica dioica*, *Areca catechu*, *Caccinia macranthera*, and *Anethum graveolens*, respectively.

**Figure 2 nanomaterials-11-02033-f002:**
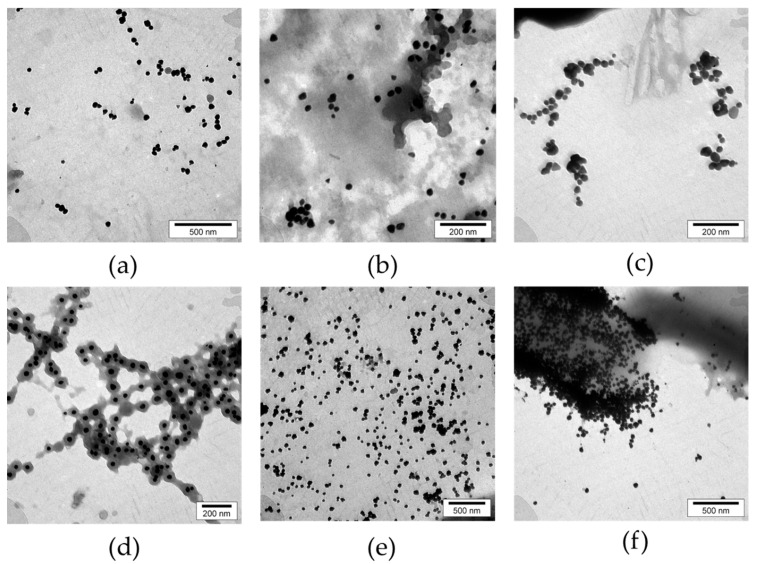
Transmission electron micrographs of six biosynthesized AuNPs by six bioactive medicinal plant extracts. Each electron micrograph is presented for its related sample. (**a**–**f**) indicated TEM microscopic evaluations of *Rosa damascena*, *Juglans regia*, *Urtica dioica*, *Areca catechu*, *Caccinia macranthera*, and *Anethum graveolens*, respectively.

**Figure 3 nanomaterials-11-02033-f003:**
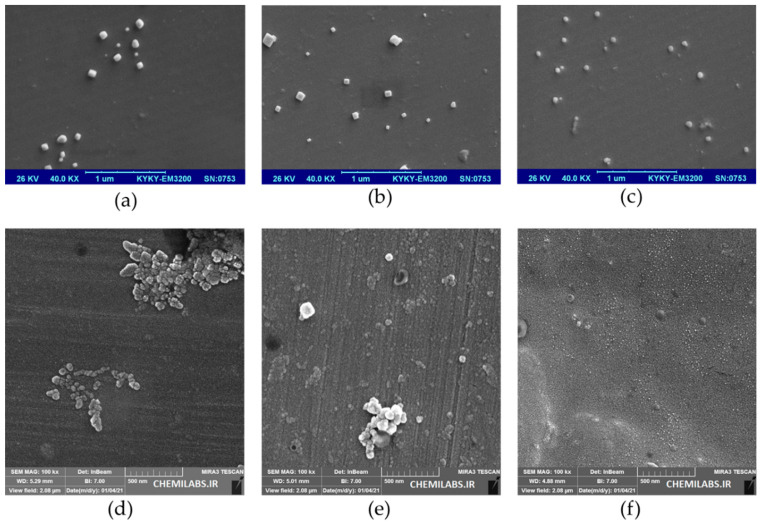
Scanning electron micrographs of biosynthesized AuNPs by six bioactive medicinal plant extracts. Each image was presented for its related sample. (**a**–**f**) indicate SEM images mediated by water extract of *Rosa damascena*, *Juglansregia*, *Caccinia macranthera*, *Urticadioica*, *Areca catechu*, and *Anethum graveolens*, respectively.

**Figure 4 nanomaterials-11-02033-f004:**
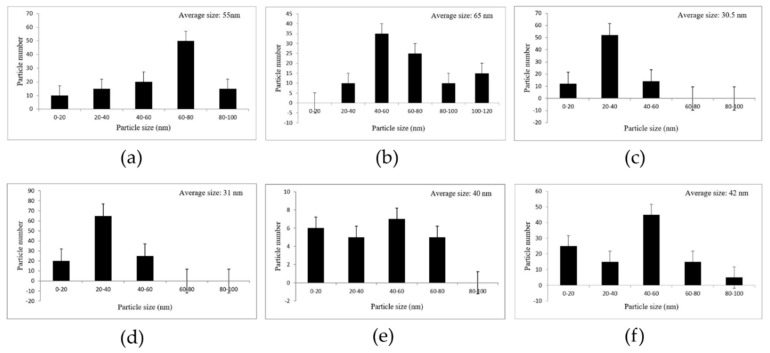
Particle size distribution histograms of the biosynthesized AuNPs via six aqueous extracts of medicinal plants used in folkloric medicine in the Middle East. Each image was presented for its related sample. (**a**–**f**) indicated the particle size distribution of *Rosa damascena*, *Juglansregia*, *Urticadioica*, *Areca catechu*, *Caccinia macranthera*, and *Anethum graveolens*, respectively.

**Figure 5 nanomaterials-11-02033-f005:**
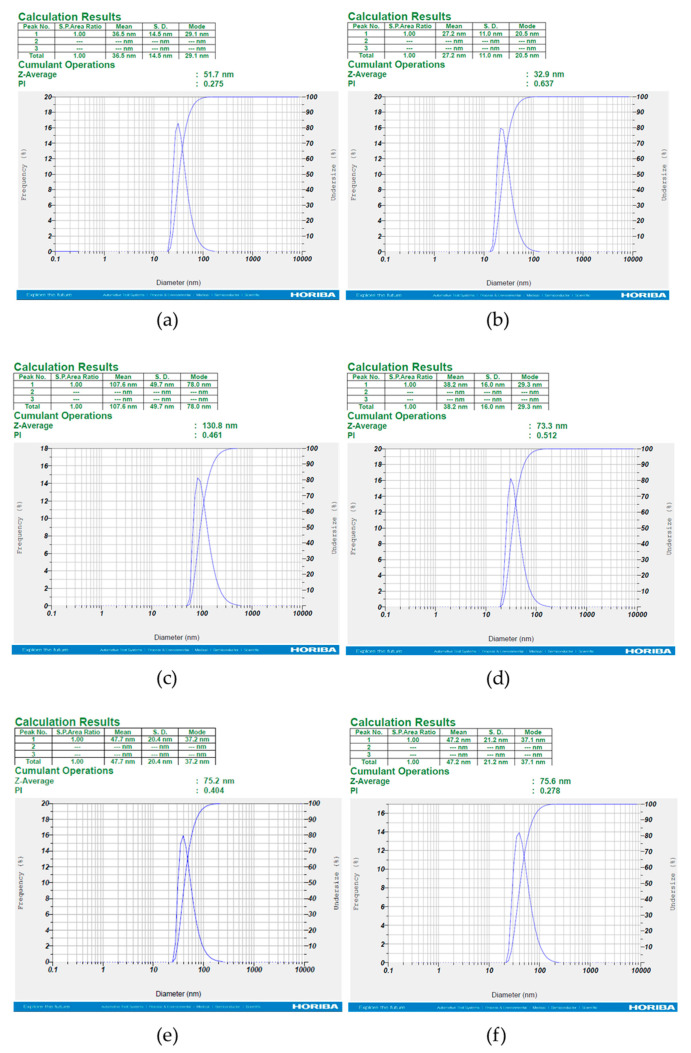
Dynamic light scattering of the biosynthesized AuNPs from water extracts of six medicinal plants. The diameters of the distribution of particles are 27.2 to 107.6 nm. The mean size distribution of AuNPs for *Rosa damascena*, *Juglans regia*, *Urtica dioica*, *Areca catechu*, *Caccinia macranthera*, and *Anethum graveolens* are shown in (**a**–**f**). The ranging of polydispersity index (PI) was between 0.27 and 0.63.

**Figure 6 nanomaterials-11-02033-f006:**
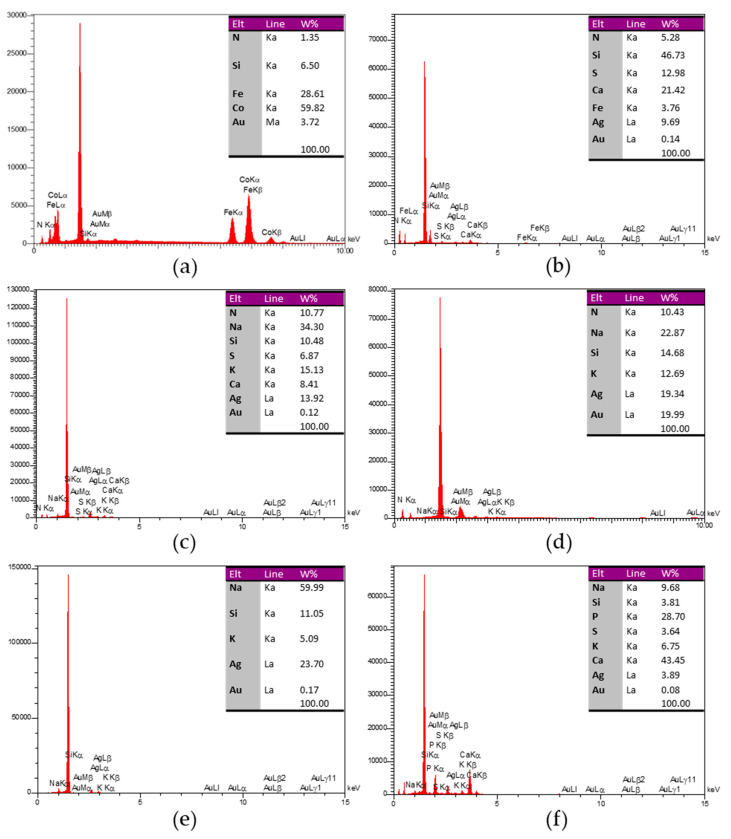
The results of energy-dispersive X-ray analysis of the biosynthesized AuNPs mediated by six water extracts of medicinal plants. Each graph and the table next to it represent its related sample. (**a**–**f**) indicate EDAX diagrams of reaction mixtures (Au^3+^, Au^0^, and the plant extract) of *Rosa damascena*, *Juglans regia*, *Caccinia macranthera*, *Urtica dioica*, *Areca catechu*, and *Anethum graveolens* respectively.

**Table 1 nanomaterials-11-02033-t001:** Newly identified plant species for the synthesis of AuNPs used in folkloric medicine.

No.	Scientific Name	Family	VN ^1^	PP ^2^
1	*Pistacialentiscus*	Anacardiaceae	ANAC120	Gu
2	*Heracleumpersicum*	Apiaceae	APIA43	Fr
3	*Artemisia cina*	Asteraceae	ASTE18	Se
4	*Pyrethrum roseum*	Asteraceae	ASTE60	Fr
5	*Echiumamoenum*	Boraginaceae	BORA23	Fl
6	*Cacciniamacranthera*	Boraginaceae	CACC64	Le
7	*Nasturtium officinalis*	Brassicaceae	BRAS49	Ab
8	*Lepidiumsativum*	Brassicaceae	BRAS66	Se
9	*Eugenia caryophyllata*	Caryophyllaceae	CARY47	Fl
10	*Fraxinus excelsior*	Fraxinaceae	FRAX111	Fr
11	*Erodium* sp.	Geraniaceae	GERA3	Ab
12	*Teucriumpolium*	Lamiaceae	LAMI24	Ab
13	*Astragalusadscendens*	Leguminosae	LEGU103	Gu
14	*Astragalus fasciculifolius*	Leguminosae	LEGU108	Gu
15	*Allium schoenoprasum*	Liliaceae	LILI92	Se
16	*Allium stipitatum*	Liliaceae	LILI8	Bu
17	*Allium schoenoprasum*	Liliaceae	LILI85	Le
18	*Sesamum indicum*	Pedaliaceae	PEDA37	Se
19	*Oryza sativa*	Poaceae	POAC82	Se
20	*Rheum ribes*	Polygonaceae	POLY65	Le
21	*Rumexalpinus*	Polygonaceae	POLY76	Fr
22	*Rheum palmatum*	Polygonaceae	POLY114	Rh
23	*Ranunculus* sp.	Ranunculaceae	RANU63	Ab
24	*Cydoniaoblonga*	Rosaceae	ROSA11	Fr
25	*Amygdaluscommunis*	Rosaceae	ROSA94	Se
26	*Prunuscerasusavium*	Rosaceae	ROSA104	Fs
27	*RubiaTinctorum*	Rubiaceae	RUBI51	Fr
28	*Valerianaofficinalis*	Valerianaceae	VALE20	Ab

^1^ VN: Voucher number of plants stored in Laboratory of Plant Systematic, College of Agriculture, Shahid Bahonar University of Kerman, Iran; ^2^ PP: Part plants used for biosynthesis of AuNPs (Ab: above-ground parts, Ba: bark, Bu: bud, Fl: flower, Fr: fruit, Fs: fruit stalk, Gl: glumes, Le: leaf, Pe: petioles, Rh: rhizome, Ro: root, Se: seeds, Sm: flower stamen and Wh: whole plant).

## Data Availability

Not applicable.

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
