# Peer review of "Environmentally Safe Biosynthesis of Gold Nanoparticles Using Plant Water Extracts"

_nanomaterials, 2021, doi:10.3390/nano11082033_

Round 1

Reviewer 1 Report

The manuscript reports the application of various plant species for the biosynthesis of gold nanoparticles. The manuscript is interesting and can be published and the following modifications:

  1. Abstract: in the third line, “survey” is not a good word. Better to be replaced by words more experimental.
  2. This is also recommended here to state the role of the plant species in the synthesis of gold nanomaterials in the abstract.
  3. Also, better to name some plants which resulted efficiently in the biosynthesis of nanomaterials.
  4. I think section 2 (materials and methods) is a bit long. Please consider shortening this section by removing some details with less importance.
  5. Perhaps table 1 can be presented as supplementary information.
  6. Also, Fig, 1 is too heavy. You may include only those which resulted in the Ag nanoparticles and transfer the rest to the supplementary information.
  7. For figure2, first, you need to make the graphs with professional curve-making programs and also to point out the position of the peaks and the respective compound(s).
  8. The authors are recommended to extend section (4) by providing more discussions about the benefits of the synthesis methods they have developed in the framework of sustainable synthesis of nanomaterials. There are some papers published recently on the sustainability of nanomaterials that can be used by the authors for more discussions and providing recommendations for future studies.

Reviewer 2 Report

The manuscript entitled “environmentally safe biosynthesis of gold nanoparticles using plant water-extracts” presents a comprehensive examination of plant extract as a medium for Au NPs synthesis. The NPs were characterized by TEM imaging as well as DLS and SEM. The authors found the NPs are ranging from 20 to 70nm, and present round shape structures. Using cell extracts for Au NPs is not novel, and the obtained NPs are not homogeneous or small enough for most applications. The EDAX results indicate that the Au NPs hold many other metal ions, e.g. cobalt iron, calcium which makes them even less attractive.

However, the nice comprehensive work may serve future researchers, therefore the presented work has some importance.

I don’t think this work suits the standards of Journal Nanomaterials, nevertheless, with some additional work the MS may suit one of the sister journals, e.g., Processes or Materials

A few comments for the authors.

  • English editing is required
  • A discussion is missing. What are the differences between the obtained NPs, analyzing the wavelengths shifts of the plasmonic absorbance could be a good start.
  • Au NPs absorb well at the visible while the biological treatment window (e.g. for cancer therapy) is lower in energy and reach near IR. Did the authors examine their samples' absorbance at the 800-900nm range? By looking at the obtained images (of the NPs) you may generate gold nanoshells or NRs that have high importance.

Round 2

Reviewer 2 Report

As requested by the reviewers, the authors thickened the discussion section. Unfortunately, the MS still requires substantial editing work.

two major points should be addressed before acceptance:

  • Scientific writing. The jargon/vocabulary used by the authors is inappropriate for a highly ranked journal like nanomaterials.
  • clear writing and grammar.

I may suggest that the authors will use professional assistance for the required editing. 

Author Response

Comments: As requested by the reviewers, the authors thickened the discussion section.
Unfortunately, the MS still requires substantial editing work.
Two major points should be addressed before acceptance:
ï‚· Scientific writing. The jargon/vocabulary used by the authors is inappropriate for a highly
ranked journal like nanomaterials.
ï‚· clear writing and grammar.
I may suggest that the authors will use professional assistance for the required editing.

Response: Thanks for satisfying our previous revision.

Related to grammatical revision of the script. We have thoroughly revised the manuscript and highlighted it with red color in the main text.

We would like to thank you once again for your consideration of our work and for inviting us to submit the revised manuscript. We look forward to hearing from you.